# Humans and LLMs rate deliberation as superior to intuition on complex reasoning tasks
Wim De Neys ✉ & Matthieu Raoelison

Influential models conceive human thinking as an interplay between intuition and deliberation. Yet, it's unclear how people actually perceive these types of reasoning. Across 13 studies ($n = 239, 241, 240, 240, 241, 240, 184, 482, 479, 240$ and $240$ for Studies 1, 2, 3, 4, 5, 6, 7, 10, 11, 12 and 13, respectively), we examined whether humans favor intuition or deliberation and if this replicates in LLMs. Participants rated individuals' reasoning quality in short vignettes that varied by reasoning type (fast-intuitive vs. slow-deliberative) and past accuracy (high, low, unspecified). Consistently, participants rated deliberative reasoning as superior to intuition, even when accounting for accuracy. Deliberative thinkers were seen as smarter and more trustworthy—a preference that held under time pressure and cognitive load, suggesting it arises intuitively. Studies with LLMs (ChatGPT 3.5 and 4) replicated the human preference pattern, indicating that AI language models capture human folk beliefs about reasoning. These findings suggest humans intuitively link deliberation with reliability and have implications for public trust in human and AI recommendations.

Popular culture often lauds the power of intuition. Albert Einstein famously characterized the intuitive mind as a "sacred gift," while Steve Jobs attributed his groundbreaking success to following his gut feeling[1,2]. This admiration for intuition is also reflected in widely publicized examples like Captain Chesley "Sully" Sullenberger, who saved his passengers by making a split-second decision to land his damaged plane on the Hudson River[3]. Popular science books such as Malcolm Gladwell's best-selling *Blink*[4] have further elevated intuitive thinking, seemingly placing it on a pedestal.

Yet, at the same time, we're all too frequently reminded to take the time to deliberate and think things through rather than to merely act on our first hunch[5–8]. This tension raises a fundamental question: How do we actually perceive intuitive versus deliberate reasoning? Do we want people to rely on intuition or deliberation when they reason? Are we more likely to trust someone's advice if they relied on their intuition, or if they took the time to think things through?

For decades, cognitive science has characterized human thinking as an interplay between fast, intuitive and slower, deliberate thought processes (or System 1 and 2, as they are often referred to [e.g., [9]]). While this research has illuminated the mechanics of both systems[10–12], we lack understanding of how people perceive and value these distinct modes of thought—what might be termed a "folk theory" of fast-and-slow thinking[13].

Although people may prefer intuition when making decisions about subjective choices (e.g., what partner to date[14,15]), some initial work does seem to suggest that humans—even from a young age[16]—generally tend to prefer deliberation over intuition when facing more objective, cognitively challenging reasoning tasks[14,17]. However, this early work often conflates thinking style and accuracy. People may prefer deliberation because they believe it will be more accurate. But what if we know that the decision maker is an experienced expert who is highly accurate? This question is particularly relevant given that celebrated cases of intuitive success—from Jobs to Sullenberger to Gladwell's examples—typically showcase experts who are consistently accurate. So, what do we prefer if two individuals are both highly accurate? Do we trust the individual who intuitively "sees" the right solution more than the one who needs to spend time and effort to arrive at an answer?

Interestingly, reasoning research has indicated that when faced with challenging brain teasers and reasoning problems, the most accurate reasoners often arrive at sounds answers intuitively[10,18–20]. Likewise, cognitive capacity seems to be more predictive of generating a sound intuition, rather than of correcting a faulty intuition through deliberation[21–23]. If our folk beliefs have picked up on this insight, there may be good reasons to prefer intuition.

The inverse scenario presents equally compelling questions: How do we perceive extensive deliberation that ultimately leads to error? Does effortful—yet unsuccessful—analysis command more respect than quick misjudgements? These questions carry profound implications for understanding public trust in our judgment, particularly crucial in an era where we face intense exposure to others' opinions and recommendations[24].

Beyond human reasoning, insight into our folk beliefs is also relevant for Artificial Intelligence development[13]. Current approaches aim to

LaPsyDE (UMR CNRS 8240), Université Paris Cité, Paris, France. ✉e-mail: wim.de-neys@u-paris.fr

enhance Large Language Models by emulating deliberate "System 2" reasoning through chain-of-thought prompts and extended computation time[2] [5–27]. If our folk beliefs indeed favour deliberation, these developments may not only improve AI accuracy but also public trust in its recommendations —potentially helping to overcome possible algorithm aversion[28].

Critically, beyond identifying our folk beliefs, it is crucial to understand their underlying nature. Specifically, if we have a preference for one reasoning mode over the other, does this preference require us to engage in deliberation or does it result from mere intuitive processing? Such insights will allow us to gauge whether our preferences are stable or if they shift depending on whether we have the time and capacity to reflect on them.

To start addressing these issues we ran 13 studies using short vignettes. Each vignette described an individual's reasoning style (either fast intuition or slow deliberation) and past accuracy (high, low, or unspecified). Participants then rated the quality of the individual's reasoning by rating, on 11-point scales, how good and smart they perceived the reasoners to be and whether they would follow their advice. Study 1 introduced this paradigm, while Studies 2–7 tested the robustness of the findings.

In Studies 8–9 we presented the vignettes to Large Lange Models (ChatGPT 3.5 and 4) to test whether they had picked up on the human preferences. Finally, Studies 10–13 tested the nature of humans' folk beliefs. We used time-pressure and load manipulations to experimentally reduce deliberate reasoning during the evaluation process. Since deliberation requires both time and cognitive resources, this allowed us to determine whether people's preference for intuition or deliberation depends itself on intuitive versus deliberate reasoning.

## Methods
### Inclusion and ethics statement
Our experimental procedures were approved by the Comité d'Ethique de la Recherche Université Paris Cité protocol 00012023-151.

### Open science and data
The research question and study design were preregistered on AsPredicted (https://aspredicted.org/). No specific analyses were preregistered. Preregistration links and dates are presented in Table 1. All data, material, preregistrations and analysis scripts can be retrieved from our OSF page: https://osf.io/6pc2e/?view_only=a5beec9676914e82baf0bbf4a10e4b64.

### Participants
All human participants were recruited on the Prolific Academic platform. They gave their informed consent prior to taking part in the studies.

### Studies 1–5 and 12–13
In each study we aimed to recruit 240 participants. This sample size allowed us to detect medium effects (0.23) between intuitive and deliberate vignettes with 95% power. Participants

were paid at a £6.00 per hour rate. They had English as their first language and held UK, USA, Ireland, Australia, Canada, or New Zealand nationality. The full sample in each study was requested to be gender balanced. Small sample deviations ($n$ +/− 1 participants) occurred due to platform-related technical issues. Full demographic sample details can be found in Table 2 and SI A.

### Studies 10–11
For the between-subject time pressure Studies 10–11 we also aimed to recruit 240 participants in each timing condition, for a total of 480 participants in each study. Other specifications (including compensation) were similar to Study 1 above. Small sample deviations ($n$ +/− 3 participants) occurred due to platform-related technical issues so we ended with 242 (deadline) + 240 (forced deliberation) participants in Study 10 and 237 (deadline) + 242 (forced deliberation) participants in Study 11. Full demographic sample details can be found in Table 2 and SI A.

### Studies 6–7
Participants in Study 6 ($n = 240$, 141 male, 94 female, 5 other/undisclosed) were of French nationality and residing in France. Participants in Study 7 were of Indian nationality ($n = 184$, 98 male, 84 female, 2 other/undisclosed) and residing in India. Because of the smaller pool of participants with these specifications on the platform, we increased the payment to a £9.00 per hour rate and dropped the gender balance requirement. As preregistered, since we did not meet the intended sample of 240 participants for Study 7 after two weeks, we closed the study and analyzed the available data. Full demographic sample details can be found in Table 2 and SI A.

### Vignettes material
Participants were presented with six, short experimental vignettes describing an individual's reasoning style (intuitive or deliberative) and past accuracy (high, low, or unspecified). To illustrate, here are examples from the intuitive-high accuracy and deliberation-low accuracy items from Study 1:

"Person A follows their intuition when reasoning about a problem. They do not spend much time or effort to arrive at a conclusion. The accuracy of their answers is very high."

"Person B reflects deeply when reasoning about a problem. They spend a lot of time and effort to arrive at a conclusion. The accuracy of their answers is very low."

In the "unspecified" vignettes the last sentence with the accuracy information was removed. Each of the vignettes was labelled with a unique letter and specified a unique combination of the reasoning mode and accuracy factor (see SI B for full vignette texts). Vignettes were presented in a counterbalanced order such that each combination of the reasoning mode and accuracy factor item would be presented as the first item an equal

## Table 1 | Preregistration information

| Study | Link | Date |
| --- | --- | --- |
| 1 | https://aspredicted.org/dw9h-cthq.pdf | 2023-12-20 |
| 2 | https://aspredicted.org/rdx8-jwcx.pdf | 2024-06-05 |
| 3 | https://aspredicted.org/9ttj-x86f.pdf | 2024-06-27 |
| 4 | https://aspredicted.org/sdh9-ghtq.pdf | 2024-02-20 |
| 5 | https://aspredicted.org/5vbb-h6r8.pdf | 2024-07-02 |
| 6 | https://aspredicted.org/gtbg-mrh4.pdf | 2024-10-31 |
| 7 | https://aspredicted.org/5mh6-mrhd.pdf | 2024-10-31 |
| 10 | https://aspredicted.org/t23j-js8q.pdf | 2024-04-30 |
| 11 | https://aspredicted.org/94mw-kpys.pdf | 2024-07-09 |
| 12 | https://aspredicted.org/nhzr-h78k.pdf | 2024-07-08 |
| 13 | https://aspredicted.org/4xts-qxxt.pdf | 2024-08-15 |

Study labels: 1 = original, 2 = replication, 3 = accuracy, 4 = implicature, 5 = 1-scale, 6 = French, 7 = Indian, 10 = timing, 11 = timing-1-scale, 12 = two-response, 13 = two-response-hard.

**Table 2 | Participants age and sex**

| Study | M | SD | n | | | |
|---|---|---|---|---|---|---|
| | | | Male | Female | Other | Total |
| 1 | 41.9 | 14.7 | 118 | 120 | 1 | 239 |
| 2 | 38.9 | 12.4 | 119 | 122 | 0 | 241 |
| 3 | 34.4 | 12.6 | 121 | 117 | 2 | 240 |
| 4 | 38.8 | 13.0 | 118 | 120 | 2 | 240 |
| 5 | 35.8 | 13.1 | 120 | 118 | 3 | 241 |
| 6 | 32.7 | 10.1 | 141 | 94 | 5 | 240 |
| 7 | 31.3 | 10.5 | 98 | 84 | 2 | 184 |
| 10 | 43.0 | 23.4 | 236 | 243 | 3 | 482 |
| 11 | 36.5 | 21.2 | 235 | 237 | 7 | 479 |
| 12 | 34.0 | 12.0 | 119 | 120 | 1 | 240 |
| 13 | 39.5 | 14.3 | 116 | 122 | 2 | 240 |

Study labels: 1 = original, 2 = replication, 3 = accuracy, 4 = implicature, 5 = 1-scale, 6 = French, 7 = Indian, 10 = timing, 11 = timing-1-scale, 12 = two-response, 13 = two-response-hard. Sex information was provided by participants by answering the following prompt: "Select your biological sex: Female/Male/Other/Prefer not to say". Other and Prefer not to say responses were collapsed together.

number of times. The presentation order of the remaining items was randomized. In addition to the six experimental vignettes we also presented an extra filler vignette with the text "There is no information about this person's profile". The general task instructions (see SI B) indicated the vignettes depicted situations where individuals encounter complex reasoning tasks.

Except for Studies 3–4 and Study 6 all studies presented the same vignettes. Study 6 adopted French translations of the English items. Study 3 used a cover story whereby we added exact numerical information to the high (95%) and low accuracy (5%) text that referred to the previous performance of the described individual (instruction text: "We will give you information about the reasoning style of different individuals. We want to know how you evaluate their reasoning skills. These people have been tested on a range of reasoning tasks and for some of them you'll be given information on their exact performance (i.e., their mean score %).") Study 4 modified the text of the highly accurate and intuitive profile to "Person A follows their intuition when reasoning about a problem. They do not need to spend much time or effort to arrive at a conclusion." to stress the efficiency of fast intuition. Full vignette text and study instructions are presented in SI B.

### Rating scales
For each vignette, participants rated the quality of the described individual's reasoning by clicking on the corresponding value on an 11-point rating sale (0–10). Except for Studies 5 and 11–13, we presented three different scales ("How good is this person at reasoning?"; "How smart is this person?"; "To what extent would you follow this person's advice about a reasoning problem?"). The scale labels were inspired by previous work measuring decision quality[17,29]. The three scales were presented simultaneously, directly under the vignette on one single page. For studies 5 and 11–13 we used a unified, single rating scale ("How good is this person at reasoning?"). End and midpoints of the scales were labelled. Higher values indicated higher perceived reasoning quality. Participants clicked on the scale number of their choice to indicate their evaluation. Participants were shown examples of the rating scales in the instructions and were given one practice trial to familiarize themselves with the response selection (see SI B for full instructions).

### Ranking question
After participants had rated the four core vignettes (i.e., high/low accuracy intuitor and deliberator), they were asked to directly rank-order them by perceived intelligence. We opted for the label "intelligence" to avoid repeating the rating scale labels. Participants saw the four vignettes and clicked on a rank order for each one of them with the following instructions: "From the descriptions below, please now rank the people from the most

intelligent (number 1) to the least intelligent (number 4). You can only place one person at each rank (i.e., no ties)." – (see SI B)

### Incentivized betting question
At the end of Study 2, participants were given the following instructions: "After this study is completed we will have four individuals with profiles matching the ones you've seen solve a reasoning problem. You will get to see the specific profiles in question again below and can bet on one of them. If the person you select solves the problem correctly, you will get a bonus payment of 1 pound. Click on the profile you'd like to bet on, then click on Next to validate." Participants then saw the four profiles they previously rank-ordered. Note that given the use of deception and the hypothetical nature of our cover story, all participants were debriefed about the nature of the study and given the bonus payment.

### LLM Studies 8–9
To test whether Large Language Models (LLMs) would capture human patterns, we used the OpenAI API to present ChatGPT models with our vignettes. We settled with ChatGPT 3.5 and ChatGPT 4 in Study 8 and 9, respectively. To best simulate human testing conditions, we ran 240 API queries in each study where we presented the vignettes in a randomized order, similarly to Study 1. We used default settings with the temperature of each model set to 1. For each query, we asked the model to rate each profile using the 3 scales. Each query was thus designed to simulate one "participant" (for further details, see SI C). These studies were not preregistered.

### Time pressure studies 10–11
Participants in Studies 10–11 were randomly allocated to either a time-pressure (restricted deliberation condition) or forced deliberation condition[30]. Study 10 used the original 3-scale format from Study 1, Study 11 implemented the single-scale format from Study 5. Response deadlines were calibrated based on the reaction times (RT) in the unconstrained Study 1 and 5. These indicated that reading the vignettes and selecting a rating on the response scales took on average 20.6 s (SD = 23.2 s) with the 3 scales in Study 1 and 12.1 s (SD = 16.3 s) with one scale in Study 5. To put participants under time-pressure we based the response deadline in Study 10 and 11 on the first quartile of the RT distribution in Study 1 (12 s) and Study 5 (6.3 s) rounded to the nearest integer, respectively (a similar approach had been previously used[31]). A few seconds before the deadline (3 s and 1.5 s, respectively) the background color of the screen turned yellow to warn participants about the upcoming deadline. Participants were reminded to respond faster on the next trials, if they missed the deadline.

In the forced deliberation condition in Study 10 and 11 participants were instructed to reflect on their choice for at least 20 s. Ratings could not advance before these 20 s had past. Note that the forced deliberation was intended as an unrestricted contrast condition for the time-pressure condition since forcing people to take more time to reflect does not imply that deliberation will be boosted per se[32,33]. The point is simply that in the forced deliberation condition, deliberation was not restricted.

Participants were given 2 practice trials to familiarize themselves with the timing conditions in which they evaluated a mock vignette. A comparison between average response times in the time-pressure conditions in Study 10 (M = 9.9 s, SD = 1.8 s) and 11 (M = 5.2 s, SD = 1 s) vs the corresponding original unconstrained Studies 1 and 5 established that participants responded significantly faster in the time-pressure conditions (t(2126.16) = 38.62, p < 0.001 and t(2135.62) = 39.80, p < 0.001, respectively).

For completion, note that for exploratory purposes we also adopted a time-pressure and forced deliberation format for the ranking question at the end of the study (with a maximal and minimal response time of 25 s and 50 s, respectively). Full study instructions can be found in SI D.

### Two-response studies 12–13
Studies 12 and 13 employed a two-response paradigm[34]. The studies adopted the single-scale format from Study 5. Participants gave two consecutive ratings to each vignette: an initial "intuitive" rating under time

pressure and cognitive load, followed by a second "deliberate" rating without restrictions.

The load task was based on previous work[35]. Before each vignette was initially presented, participants briefly saw a complex visual matrix pattern for 2 s (see SI E). After participants had entered their initial vignette rating, they needed to select the to-be-remembered matrix among four candidates (see SI E). Afterward, the vignette was presented again and participants could take all the time they wanted to reflect on their final evaluation without any constraints.

In Study 12 the initial response deadline was 6 s (as in Study 11) and the memorization concerned a 4-cross pattern in a $3 \times 3$ grid. Given there is no absolute established timing or load deliberation threshold[10], following previous work[34], we tried to further limit the theoretical possibility for deliberation during the initial response stage by further restricting the deadline (5.5 s) and increasing the load (5-cross pattern in a $4 \times 4$ grid, see SI E) in Study 13.

1.5 s before the deadline the background color of the screen turned yellow to warn participants about the upcoming deadline. Participants were reminded to respond faster on the next trials or pay more attention to memorizing the load pattern, if they missed the deadline or failed the load. Participants were familiarized with the two-response procedure in 2 practice trials with mock vignettes. Full study instructions can be found in SI E.

### Trial exclusions for time-pressure and two-response studies 10–13

In line with our preregistration, we discarded data from one participant who missed the deadline on more than 2 trials in Study 10 and further discarded 45 individual missing trials (i.e., trials with more than 1 rating scale value out of 3 missing; out of 3367, i.e., 1.3% of trials). We also discarded data from two participants who missed more than 2 trials in Study 11. We replaced single missing values with sample means in Study 10 in 80 trials (out of 3367, i.e., 2.4% of trials) and did the same in Study 11 for 88 trials (out of 3339, i.e., 2.6% of trials). See SI D for details.

In Study 12, we discarded data from 5 participants who missed the deadline on more than 2 trials and/or who failed the load task in more than half of their trials and we similarly discarded data from 43 participants in Study 13. We replaced single missing values with sample means in Study 12 in 100 trials (out of 1645, i.e., 6.1% of trials) and did the same in Study 13 for 77 trials (out of 1379, i.e., 5.6% of trials). See SI E for details.

### Statistical analysis

Data distribution for ANOVAs was assumed to be normal but this was not formally tested. Note that whenever Mauchly's test indicated that the assumption of sphericity had been violated, we report Greenhouse–Geisser corrected ANOVA results. All analyses (all two-sided) were done using the following R packages (in alphabetical order): *afex*[36], *broom*[37], *effectsize*[38], *emmeans*[39], *ginnards*[40], *ggpubr*[41], *ggrepel*[42], *gt*[43], *here*[44], *janitor*[45], *knitr*[46], *openai*[47], *patchwork*[48], *rstatix*[49], *scales*[50], *tidyverse*[51]. Computation of Cohen's *d* for contrasts and additional Bayesian analyses were run in JASP[52].

### Results
### Study 1: General preference for deliberation over intuition

Correlations among our three rating scales were consistently high (Study 1: $0.79 \leq rs \leq 0.87$; all other studies $0.78 \leq rs \leq 0.91$, see SI F) and were combined into a single composite preference scale reflecting the overall perceived reasoning quality. Figure 1 shows the results. As indicated in Table 3, a 3 (Accuracy Information: High, Low, Unspecified) x 2 (Reasoning Mode: Intuition, Deliberation) within-subjects ANOVA on the composite rating showed there was a main effect of Accuracy information, $F(1.65, 392.91) = 1264.5$, $p < 0.001$, $\eta_p^2 = 0.84$ [0.82, 1.00]. Participants preferred reasoners with high accuracy over those with low accuracy, with the unspecified condition in between, indicating that the accuracy manipulation was effective as intended.

More critically, there was a main effect of Reasoning mode, $F(1, 238) = 348.6$, $p < 0.001$, $\eta_p^2 = 0.59$ [0.53, 1.00], and a Reasoning mode x Accuracy interaction, $F(2, 476) = 80.47$, $p < 0.001$, $\eta_p^2 = 0.25$ [0.20, 1.00], indicating a general preference for the deliberate over the intuitive reasoner

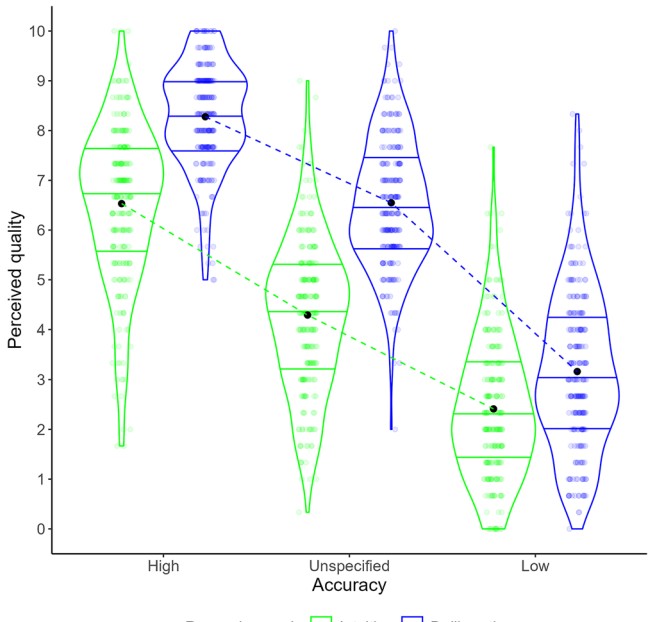

**Fig. 1 | Perceived reasoning quality as a function of an individual's portrayed past accuracy and reasoning mode in Study 1.** Violin plots showing the overall perceived reasoning quality averaged across 3 rating scales (i.e., "how good is this person at reasoning?"; "how smart is this reasoner?"; "to what extent would you follow their advice?" – see SI B) for $n = 239$ participants. Black dots indicate the average. Horizontal lines represent the 25th, 50th and 75th percentiles. Unsurprisingly, individuals described as having higher past accuracy are preferred over individuals with lower (or unspecified) accuracy. Key result is that individuals who are portrayed as adopting a deliberative reasoning mode (blue) are consistently rated as superior to individuals portrayed as being more intuitive (green), irrespective of past accuracy.

that was strongest when accuracy information was absent and slightly attenuated in the high- and low-accuracy conditions. Importantly, the preference for deliberation never disappeared or reversed and remained significant across all accuracy conditions ($7.6 \leq ts \leq 19.3$, $4 \times 10^{-50} \leq ps \leq 3.9 \times 10^{-13}$, $0.53 \leq ds \leq 1.58$, see Supplementary Table 6 in SI G). These findings thus indicate a general preference for deliberation over intuition, regardless of accuracy.

To check for potential bias from the within-subject design, we also ran a between-subjects analysis on the first item each participant evaluated, yielding similar results (see SI H). For further validation, after participants had rated all vignettes, they were also asked to directly rank-order four core vignettes (i.e., high/low accuracy intuitor and deliberator) by perceived reasoning quality. Average ranking scores confirmed the rating scale conclusions, with deliberate reasoners ranked higher than intuitive ones (see SI I).

### Study 2–7: Robust preference for deliberation

Study 2–7 served as control studies to validate the robustness of the findings and rule out alternative explanations. Study 2 concerned a simple direct Study 1 replication, while Study 3 and 4 altered vignette content to address specific concerns. One concern was that our verbal accuracy label (e.g., "The accuracy of their answers is very high") might still allow participants to infer that deliberation implicitly yields higher accuracy than intuition, artificially boosting deliberation ratings. To counter this, Study 3 presented a cover story with exact numerical scores on a reasoning test (e.g., "The accuracy of their answers is very high (95%)"), with identical scores for both intuitive and deliberative vignettes.

Another concern was that the intuitive vignette wording ("They do not spend much time or effort to arrive at a conclusion") might skew evaluations negatively by not sufficiently emphasizing the efficiency of fast intuition. In Study 4, we adapted the phrasing to emphasize efficiency (e.g., "They do not *need to* spend much time or effort to arrive at a conclusion") following

**Table 3 | Within-subject ANOVA (and sphericity) table for Studies 1–7**

| Study | Effect | ANOVA | | | | | Sphericity | | |
|---|---|---|---|---|---|---|---|---|---|
| | | DFn | DFd | F | P | $\eta_p^2$ | W | p | $\hat{\varepsilon}$ |
| 1 | Mode | 1.00 | 238.00 | 348.61 | <0.001 | 0.594 [0.53, 1.00] | | | |
| | Accuracy | 1.65 | 392.91 | 1264.48 | <0.001 | 0.842 [0.82, 1.00] | 0.79 | <0.001 | 0.83 |
| | Mode:Accuracy | 2.00 | 476.00 | 80.47 | <0.001 | 0.253 [0.20, 1.00] | 0.98 | 0.146 | 0.98 |
| 2 | Mode | 1.00 | 240.00 | 381.65 | <0.001 | 0.614 [0.56, 1.00] | | | |
| | Accuracy | 1.58 | 378.67 | 1293.51 | <0.001 | 0.843 [0.83, 1.00] | 0.73 | <0.001 | 0.79 |
| | Mode:Accuracy | 2.00 | 480.00 | 71.05 | <0.001 | 0.228 [0.18, 1.00] | 0.99 | 0.228 | 0.99 |
| 3 | Mode | 1.00 | 239.00 | 314.94 | <0.001 | 0.569 [0.50, 1.00] | | | |
| | Accuracy | 1.74 | 415.19 | 1620.74 | <0.001 | 0.871 [0.86, 1.00] | 0.85 | <0.001 | 0.87 |
| | Mode:Accuracy | 2.00 | 478.00 | 89.07 | <0.001 | 0.272 [0.22, 1.00] | 0.98 | 0.053 | 0.98 |
| 4 | Mode | 1.00 | 239.00 | 310.60 | <0.001 | 0.565 [0.50, 1.00] | | | |
| | Accuracy | 1.60 | 382.89 | 1040.62 | <0.001 | 0.813 [0.79, 1.00] | 0.75 | <0.001 | 0.8 |
| | Mode:Accuracy | 2.00 | 478.00 | 72.17 | <0.001 | 0.232 [0.18, 1.00] | 1 | 0.961 | 1 |
| 5 | Mode | 1.00 | 240.00 | 346.58 | <0.001 | 0.591 [0.53, 1.00] | | | |
| | Accuracy | 1.57 | 377.82 | 949.50 | <0.001 | 0.798 [0.78, 1.00] | 0.73 | <0.001 | 0.79 |
| | Mode:Accuracy | 1.93 | 462.57 | 56.87 | <0.001 | 0.192 [0.14, 1.00] | 0.96 | 0.010 | 0.96 |
| 6 | Mode | 1.00 | 239.00 | 160.54 | <0.001 | 0.402 [.33, 1.00] | | | |
| | Accuracy | 1.65 | 395.29 | 1199.43 | <0.001 | 0.834 [.81, 1.00] | 0.79 | <0.001 | 0.83 |
| | Mode:Accuracy | 2.00 | 478.00 | 46.16 | <0.001 | 0.162 [.11, 1.00] | 0.99 | 0.488 | 0.99 |
| 7 | Mode | 1.00 | 183.00 | 111.31 | <0.001 | 0.378 [.29, 1.00] | | | |
| | Accuracy | 1.60 | 292.68 | 498.97 | <0.001 | 0.732 [.70, 1.00] | 0.75 | <0.001 | 0.8 |
| | Mode:Accuracy | 2.00 | 366.00 | 27.94 | <0.001 | 0.132 [0.08, 1.00] | 0.98 | 0.158 | 0.98 |

Study labels: 1 = original (n = 239), 2 = replication (n = 241), 3 = accuracy (n = 240), 4 = implicature (n = 240), 5 = 1-scale (n = 241), 6 = French (n = 240), 7 = Indian (n = 184). The Greenhouse-Geisser correction is only applied to df when the sphericity assumption is violated. Square brackets indicate 95% CI.

pragmatic implicature principles[53]. Study 5 addressed potential bias from asking participants for three consecutive ratings by adopting a single evaluation scale for overall reasoning quality.

Participants in Studies 1–5 were all native English speakers living in anglophone, Western countries. Studies 6–7 tested for minimal generalization of the results across languages and cultures. Study 6 tested French-speaking participants in continental Europe, while Study 7 tested Indian nationals living in India. ANOVA results are reported in Table 3.

As shown in Fig. 2A, all studies replicated the primary pattern from Study 1 (see SI G for details). Although accuracy information attenuated the preference, participants consistently favoured deliberation over intuition in all conditions ($4.19 \le ts \le 20.42$, $6.5 \times 10^{-54} \le ps \le 4.3 \times 10^{-5}$, $0.26 \le ds \le 1.69$, see Supplementary Table 6 in SI G).

### Incentivized preference test

At the end of replication Study 2 we also tested whether participants' verbal preference was consequential. Although participants verbally rated deliberation more highly, this preference could potentially stem from social desirability rather than genuine conviction. To test this, we used an incentivized paradigm where participants could double their participation fee by betting on the performance of either a highly accurate intuitor or deliberator (as well as a lowly accurate intuitor and deliberator, for control purposes). A cover story explained that each individual had solved a reasoning test problem, and participants could bet on one of them, receiving a £1 bonus if the individual's answer was correct. In line with the rating findings, participants were far more likely to bet on the highly accurate deliberator (76.8%) than on the intuitor (21.6%), $\chi^2(3) = 372$, $p < 0.001$.

Interestingly, before the betting question participants also rank ordered the vignettes (unincentivized, see Study 1). Of the participants who ranked the highly accurate deliberator higher than the intuitor, 86.6% also betted on the deliberator. However, among those whose ranking showed a preference for the intuitor, only 46.4% opted for it when money was at stake in the betting

question (see SI J). Thus, when choices had real monetary consequences, the preference for deliberation over intuition became even more pronounced.

### Study 8–9: Large language models reflect human preferences

In Studies 8–9, we tested whether Large Language Models (LLMs), specifically ChatGPT 3.5 and 4, would mirror the human preference pattern. Using our original vignettes and rating questions, we ran 240 API queries with each model to simulate the same number of subjects as in our individual previous studies. Figure 2B displays the LLM results alongside the aggregate human data from Studies 1–7.

As shown in Fig. 2B, both ChatGPT 3.5 and 4 produced response patterns strikingly similar to those of human participants: a consistent preference for deliberation over intuition, strongest when accuracy information was absent and slightly reduced but still significant in the high- and low-accuracy conditions ($17.02 \le ts \le 56.97$, $8.6 \times 10^{-135} \le ps \le 6.4 \times 10^{-43}$, $1.44 \le ds \le 4.77$; see Supplementary Table 13 in SI K for details). Notably, the absolute ratings in each condition closely aligned with human responses, with condition means differing on average by 0.62 points for ChatGPT 3.5 and 0.44 points for ChatGPT 4 on the 11-point scale—representing less than 7% deviation from human ratings in Study 1 (see SI K).

These results suggest that LLMs like ChatGPT have captured human preferences for deliberation over intuition, likely because these patterns are well-represented in the language data used to train the models. This alignment underscores that the preference for deliberative reasoning may be a widely shared belief evident in common language and communication.

### Study 10–13: Intuitive nature of deliberation preference

Since Studies 1–9 consistently showed a preference for deliberation over intuition, Studies 10–13 examined the cognitive nature of this preference. One possibility is that the preference for deliberation arises intuitively, without requiring reflection. Alternatively, it might be that individuals actually have an initial intuitive preference for intuition, which shifts towards deliberation

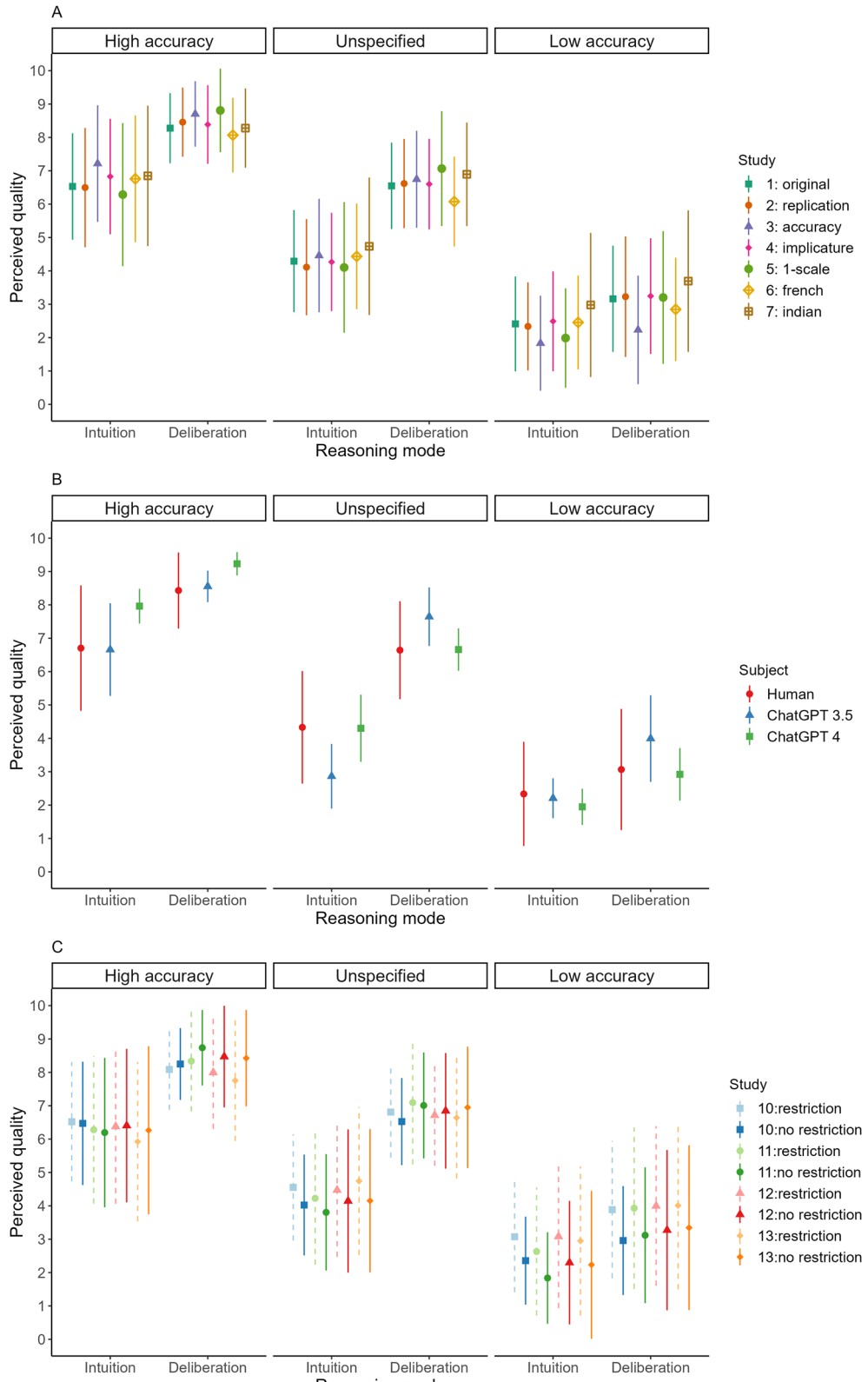

when given time to reflect. To test this, we used ever more challenging time-pressure and load manipulations to experimentally reduce deliberate reasoning during the evaluation process in Studies 10–13. If the deliberation preference is itself intuitive, we should observe it even when deliberation is minimized. Conversely, if it requires deliberation, the preference should disappear or reverse when participants are forced to rely on intuition.

In Study 10, we used the original 3-scale format from Study 1, with one group of participants completing evaluations under time pressure (12 s deadline) and a second group under a forced deliberation condition[30], which required participants to reflect for at least 20 s before responding. Study 11 implemented a similar design using the single-scale format from Study 5, allowing for a more rigorous time constraint (6 s deadline). Studies 12 and

**Fig. 2 | Perceived reasoning quality as a function of an individual's portrayed past accuracy and reasoning mode across all studies.** Panel (**A**) shows the results for Studies 1–7 ($n = 239, 241, 240, 240, 241, 240$ and $184$, respectively) indicating that the general preference for deliberation over intuition is robust and replicated in all control studies. Panel (**B**) shows that Large Language Models (Study 8: ChatGPT 3.5; Study 9: ChatGPT 4; $n = 240$ each) reflect the human preferences and also consistently favor deliberation over intuition. Results under the "Human" label concern the data from participants in Study 1–7 ($n = 1625$). Panel (**C**) shows the results from Studies 10–13 that experimentally restricted deliberation during the evaluation

process. Study labels: 10 = timing ($n = 241$ and $240$ with and without restriction, respectively), 11 = timing-1-scale ($n = 236$ and $241$ with and without restriction, respectively), 12 = two-response ($n = 235$), 13 = two-response-hard ($n = 197$). Light-colored dashed lines indicate conditions where deliberation was experimentally restricted, while dark-colored solid lines indicate unrestricted conditions. Results indicate that the general preference for deliberation over intuition was still observed under constraints suggesting it is primarily intuitive in nature. Note that for studies adopting multiple rating scales (Study 1–4, Study 6–10), displayed results again concern the overall average. Error bars indicate SDs, dots indicate the average.

## Table 4 | Within-subject ANOVA (and sphericity) table for Studies 10–13

| Study | Effect | DFn | ANOVA | | | | | Sphericity | | |
| | | | DFd | $F$ | $p$ | $BF_{inclusion}$ | $\eta_p^2$s | W | $p$ | $\hat{\varepsilon}$ |
|---|---|---|---|---|---|---|---|---|---|---|
| 10 | Condition | 1.00 | 446.00 | 19.55 | <0.001 | $9.75 \times 10^6$ | 0.04 [0.02, 1.00] | | | |
| | Mode | 1.00 | 446.00 | 637.59 | <0.001 | Infinity | 0.59 [0.54, 1.00] | | | |
| | Accuracy | 1.61 | 717.83 | 1661.59 | <0.001 | Infinity | 0.79 [0.77, 1.00] | 0.76 | <0.001 | 0.8 |
| | Condition:Mode | 1.00 | 446.00 | 0.14 | 0.704 | 0.22 | <0.01 [0.00, 1.00] | | | |
| | Condition: Accuracy | 1.61 | 717.83 | 15.06 | <0.001 | $6.56 \times 10^4$ | 0.03 [0.02, 1.00] | 0.76 | <0.001 | 0.8 |
| | Mode:Accuracy | 2.00 | 892.00 | 145.20 | <0.001 | Infinity | 0.25 [0.21, 1.00] | 1 | 0.612 | 1 |
| | Condition:Mode:Accuracy | 2.00 | 892.00 | 2 | 0.136 | 0.17 | <0.01 [0.00, 1.00] | 1 | 0.612 | 1 |
| 11 | Condition | 1.00 | 475.00 | 12.62 | <0.001 | $1.25 \times 10^6$ | 0.03 [0.01, 1.00] | | | |
| | Mode | 1.00 | 475.00 | 667.68 | <0.001 | Infinity | 0.58 [0.54, 1.00] | | | |
| | Accuracy | 1.73 | 819.97 | 1446.57 | <0.001 | Infinity | 0.75 [0.73, 1.00] | 0.84 | <0.001 | 0.86 |
| | Condition:Mode | 1.00 | 475.00 | 2.45 | 0.118 | 0.67 | <0.01 [0.00, 1.00] | | | |
| | Condition: Accuracy | 1.73 | 819.97 | 16.38 | <0.001 | $1.97 \times 10^5$ | 0.03 [0.02, 1.00] | 0.84 | <0.001 | 0.86 |
| | Mode:Accuracy | 2.00 | 950.00 | 94.68 | <0.001 | Infinity | 0.17 [0.13, 1.00] | 1 | 0.377 | 1 |
| | Condition:Mode:Accuracy | 2.00 | 950.00 | 2.05 | 0.129 | 0.42 | <0.01 [0.00, 1.00] | 1 | 0.377 | 1 |
| 12 | Response | 1.00 | 234.00 | 15.36 | <0.001 | Infinity | 0.06 [0.02, 1.00] | | | |
| | Mode | 1.00 | 234.00 | 226.42 | <0.001 | Infinity | 0.49 [0.42, 1.00] | | | |
| | Accuracy | 1.48 | 345.39 | 591.65 | <0.001 | Infinity | 0.72 [0.68, 1.00] | 0.65 | <0.001 | 0.74 |
| | Response:Mode | 1.00 | 234.00 | 9.62 | 0.002 | 12.84 | 0.04 [0.01, 1.00] | | | |
| | Response: Accuracy | 2.00 | 468.00 | 35.69 | <0.001 | $1.02 \times 10^{12}$ | 0.13 [0.09, 1.00] | 0.98 | 0.076 | 0.98 |
| | Mode:Accuracy | 1.94 | 453.08 | 41.43 | <0.001 | Infinity | 0.15 [0.10, 1.00] | 0.97 | 0.020 | 0.97 |
| | Response:Mode:Accuracy | 2.00 | 468.00 | 1.91 | 0.15 | 2.11 | <0.01 [0.00, 1.00] | 0.99 | 0.230 | 0.99 |
| 13 | Response | 1.00 | 196.00 | 2.75 | 0.099 | $3.29 \times 10^{11}$ | 0.01 [0.00, 1.00] | | | |
| | Mode | 1.00 | 196.00 | 274.82 | <0.001 | $1.29 \times 10^{13}$ | 0.58 [0.51, 1.00] | | | |
| | Accuracy | 1.63 | 318.95 | 390.23 | <0.001 | $1.29 \times 10^{13}$ | 0.67 [0.63, 1.00] | 0.77 | <0.001 | 0.81 |
| | Response:Mode | 1.00 | 196.00 | 11.70 | <0.001 | 86.09 | 0.06 [0.02, 1.00] | | | |
| | Response: Accuracy | 2.00 | 392.00 | 38.10 | <0.001 | $6.85 \times 10^{11}$ | 0.16 [0.11, 1.00] | 0.97 | 0.083 | 0.98 |
| | Mode:Accuracy | 2.00 | 392.00 | 19.99 | <0.001 | $1.28 \times 10^7$ | 0.09 [0.05, 1.00] | 1 | 0.63 | 1 |
| | Response:Mode:Accuracy | 2.00 | 392.00 | 4.07 | 0.018 | 39.84 | 0.02 [0.00, 1.00] | 1 | 0.621 | 1 |

Study labels: 10 = timing ($n = 241$ and $240$ with and without restriction, respectively), 11 = timing-1-scale ($n = 236$ and $241$ with and without restriction, respectively), 12 = two-response ($n = 235$), 13 = two-response-hard ($n = 197$). The Greenhouse-Geisser correction is only applied to df when the sphericity assumption is violated. The $BF_{inclusion}$ column indicates the inclusion Bayes factor for each predictor based on model-averaged results from Bayesian ANOVAs run in JASP with default priors. Higher values indicate increasing evidence in favor of an effect of the predictor. The reciprocal $BF_{01}$ indicates the related support for the null hypothesis. Square brackets indicate 95% CI.

13 employed a two-response paradigm[18,54], in which participants gave an initial "intuitive" single-scale rating under time pressure and cognitive load, followed by a second "deliberate" single-scale rating without restrictions. In the cognitive load task, participants memorized complex visual patterns (4 crosses in a $3 \times 3$ matrix in Study 12, increasing to 5 crosses in a $4 \times 4$ matrix in Study 13[34],). Time-pressure durations were calibrated based on the vignette reading and rating response times in the unconstrained deliberation Studies 1 and 5 (see Methods), with Study 13 incorporating a further reduction (5.5 s deadline) based on initial response times in Study 12.

Figure 2C presents the results. In each study, light-colored dashed lines indicate conditions where deliberation was experimentally restricted, while dark-colored solid lines indicate unrestricted conditions. Visual inspection

of Fig. 2C shows that restricting deliberation had minimal impact on the preference for deliberation: participants consistently preferred deliberation over intuition, even under significant constraints, closely matching the patterns seen in Studies 1–7 and the unrestricted conditions in Studies 10–13. This suggests that the preference for deliberation does not depend on deliberate reasoning but is primarily intuitive.

For statistical analysis, we conducted 3 (Vignette Accuracy Information: High, Low, Unspecified) x 2 (Reasoning Mode: Intuition, Deliberation) x 2 (Deliberation Restriction: Restricted, Unrestricted) ANOVAs for each study. Deliberation restriction was a between-subjects factor in Studies 10–11 and within-subjects factor in Studies 12–13. Table 4 presents the results. All studies showed main effects for Accuracy ($390.23 \leq Fs \leq 1661.59$,

$5.9 \times 10^{-250} \leq ps \leq 4.8 \times 10^{-77}$, $0.67 \leq \eta_p^2 \leq 0.79$) and Reasoning Mode ($226.42 \leq Fs \leq 667.68$, $1.37 \times 10^{-92} \leq ps \leq 3.01 \times 10^{-36}$, $0.49 \leq \eta_p^2 \leq 0.59$), as well as an Accuracy x Reasoning Mode interaction ($19.99 \leq Fs \leq 145.20$, $2.58 \times 10^{-55} \leq ps \leq 5.43 \times 10^{-9}$, $0.09 \leq \eta_p^2 \leq 0.25$), consistent with Studies 1–9. Crucially, in all accuracy conditions, participants consistently preferred deliberation over intuition across both restricted and unrestricted conditions ($5.04 \leq ts \leq 19.78$, $3.1 \times 10^{-62} \leq ps \leq 9.3 \times 10^{-7}$, $0.37 \leq ds \leq 1.72$, see Supplementary Table 16 in SI L).

ANOVA results further revealed (marginally) significant main effects for Deliberation Restriction ($2.75 \leq Fs \leq 19.55$, $1.23 \times 10^{-5} \leq ps \leq 0.099$, $0.01 \leq \eta_p^2 \leq 0.06$) and a Restriction x Accuracy interaction ($15.06 \leq Fs \leq 38.10$, $7.55 \times 10^{-16} \leq ps < 3.5 \times 10^{-6}$, $0.03 \leq \eta_p^2 \leq 0.16$), indicating that unrestricted deliberation led to generally lower ratings, particularly in the low-accuracy condition. In Studies 12 and 13, we also found a Restriction x Reasoning Mode interaction ($9.62 \leq Fs \leq 11.70$, $7.59 \times 10^{-4} \leq ps \leq .002$, $0.04 \leq \eta_p^2 \leq 0.06$), with a three-way interaction observed in Study 13 ($F(2, 392) = 4.072$, $p = 0.018$, $\eta_p^2 = 0.02$ [0.00, 1.00]). Here, lower ratings post-deliberation tended to be slightly more pronounced for intuitive vignettes, particularly in the absence of accuracy information. In Study 13's unspecified accuracy condition, for example, the deliberation preference gap narrowed from 2.80 points ($SD = 2.98$) in the unrestricted condition to 1.89 points ($SD = 2.77$) in the restricted condition. Nonetheless, even in this specific case, the deliberation preference remained robust, neither disappearing nor reversing ($t = 9.59$, $p < .001$, $d = 0.88$ [0.68, 1.08]). Together, these results support the conclusion that the preference for deliberation over intuition is itself predominantly driven by intuitive processing.

For clarity, we should stress that—as indicated above—some of the ANOVA results involving the restriction factor and its interactions yielded null effects. In line with the journal's guidelines and reviewer suggestions, we also ran Bayes Factor analyses. These indicate that support for the null hypothesis concerning the restriction effects is relatively weak ($0 \leq BF_{01} \leq 5.88$, see SI L). That said, our central claim does not rest on the absence of restriction effects in the ANOVA analyses per se. Rather, the key finding is that, across all restriction conditions, participants showed a significant pairwise preference for deliberation over intuition, regardless of accuracy. This indicates that the deliberation preference is robust across our accuracy and restriction manipulations. For completeness, we also ran a Bayes Factors analysis for the pairwise contrasts between intuitive and deliberative reasoning mode vignettes. Results consistently showed extreme evidence in favor of a deliberation preference across all restricted conditions ($1.1 \times 10^4 < BF_{10} < 1.22 \times 10^{51}$, see SI L). This further supports the conclusion that the preference for deliberation does not itself require deliberative processing.

## Discussion

Across 13 studies, we find a robust preference for deliberative over intuitive reasoning, even when accuracy is controlled and participants are under time pressure or cognitive load. Although intuition is often celebrated in popular culture, people seem to readily regard deliberation as a marker of good, thoughtful reasoning. Interestingly, large language models (LLMs) like ChatGPT 3.5 and 4 showed a similar preference, suggesting that the association between deliberation and quality is deeply embedded in human language and communication.

Our results help clarify people's folk theory about intuitive and deliberate reasoning: while fast-and-slow dual-process models[9] are widely known, few studies have directly examined these folk beliefs. The robust preference for deliberation we observed aligns with early research suggesting that people favour deliberation for complex cognitive tasks[14,16,17]. Our findings refine these insights by demonstrating that people's appreciation for deliberation is largely intuitive and independent of accuracy. Critically, our cognitive strain manipulations shed light on the underlying cognitive mechanism driving the deliberation preference, suggesting that it operates as an intuitive heuristic rather than a product of extended reflection.

This intuitive link between deliberation and perceived reliability has significant broader implications, as it suggests that public trust in decision-making could be readily cultivated, but it also runs the risk of exploitation[55]. For example, in contexts such as expert advice, legal rulings, and educational settings, the public's intuitive link between deliberation and careful reasoning could shape confidence in institutional decision-making. Policy-makers, healthcare providers, and legal authorities, for instance, might enhance public trust by emphasizing deliberative practices, even in cases where intuitive insights may yield comparably accurate outcomes[56,57].

This further extends to the realm of artificial intelligence, where preference patterns mirrored by LLMs (such as ChatGPT 3.5 and 4) suggest that these models have encoded human tendencies to favor deliberative reasoning. Given current advancements aimed at enabling models to emulate "System 2" reasoning (e.g., through chain-of-thought prompts or increased inference compute time[25–27,58]), our results support the notion that deliberation-aligned outputs may foster public trust in AI-driven recommendations. For high-stakes applications, such as medical diagnostics or legal analysis, AI systems that transparently mimic deliberative reasoning may not only enhance accuracy but also alleviate algorithm aversion by aligning with users' reasoning preferences. However, this preference for deliberative reasoning also highlights a risk: both AI and human advisors might unduly increase trust in their recommendations by strategically presenting information as deliberative, regardless of actual quality or accuracy. Consequently, while deliberation-aligned outputs may enhance trust, there may also be a pressing need for safeguards against its potential misuse.

## Limitations

When considering the broader implications of our findings, it is essential to acknowledge certain limitations of our studies. First, we focused specifically on folk beliefs about the role of intuition and deliberation in addressing cognitively challenging problems. Our vignettes were designed to depict situations where individuals encounter complex reasoning tasks. In trivial or straightforward scenarios (e.g., answering, "How much is 5 + 5?"), people are unlikely to favor a reasoner who needs slow, painstaking deliberation; in such cases, the need to deliberate could signal cognitive limitations rather than sophistication[17]. Similarly, for non-cognitive or subjective judgments (e.g., a partner asking, "Do you love me?"), people may prefer a swift, intuitive response, which is often seen as more authentic[14,15,59]. Thus, it is improbable that our findings extend to (AI) recommendations for highly subjective areas like romantic choices or personal preferences in art or music, for example. More generally, it would be beneficial to investigate the generalizability of these preferences across diverse judgment domains.

Second, while our vignette-based approach allowed us to isolate reasoning mode as a variable, it also comes with limitations. Vignettes, though controlled, do not capture the full ecological complexity of real-world judgments, where factors like social context and personal experience could influence reasoning preferences. Moreover, although our incentivized paradigm suggests a genuine preference for deliberation over intuition, future research could confirm these findings in more interactive settings. For instance, if individuals repeatedly interact with a person and learn from these experiences, their initial preference, based on a single descriptive vignette, might shift. Clearly, given that people often make trust decisions on initial impressions, especially in first-time interactions, our approach still captures an essential aspect of real-world judgment. At the same time, it would be valuable to further explore how preferences for deliberation evolve with ongoing experience and familiarity in future work.

Third, our studies asked participants to evaluate the inferences of a person who used a specific reasoning style (intuition vs. deliberation). A further question is whether people also believe these styles can shift—do they think that a reasoner can switch from one style to another? Put differently, is reasoning style seen as a stable trait or as something more flexible and context-dependent? Our studies did not address this issue. Our goal was to assess how people evaluate a reasoner based on the style used in a specific moment. The belief that reasoning styles may shift over time does not undermine such evaluations. Even if a person might reason differently in the future, people can still judge the quality of their reasoning in the situation at

hand. Nevertheless, although the malleability question was orthogonal to our central aim, it presents an interesting avenue for future research.

Finally, although we tested for minimal generalization across languages and cultures, our sample was limited to a relatively highly educated population from industrialized countries. Future studies could extend this work to encompass broader educational, cultural, and geographic backgrounds to assess the universality of the findings.

In conclusion, our findings reveal a robust, intuitive preference for deliberative reasoning that is echoed in AI models like LLMs. This highlights an implicit belief associating deliberation with reliable, thoughtful reasoning, carrying implications for fostering public trust while also presenting risks of exploitation. At the same time, the findings are shaped by certain boundary conditions—including our focus on cognitively challenging tasks, vignette-based methods, and relatively educated samples. Future work will be needed to explore how robust these preferences are across different judgment domains, real-world settings, and more diverse populations.

## Data availability

All data are available on our OSF page: https://osf.io/6pc2e/?view_only= a5beec9676914e82baf0bbf4a10e4b64.

## Code availability

All analysis scripts are available on our OSF page: https://osf.io/6pc2e/? view_only=a5beec9676914e82baf0bbf4a10e4b64.

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

## Acknowledgements

Preparation of this manuscript benefitted from support from the European Union under Horizon Europe Programme Grant Agreement no. 101120763 – TANGO. Views and opinions expressed are however those of the authors only and do not necessarily reflect those of the European Union or the European Health and Digital Executive Agency (HaDEA). Neither the European Union nor the granting authority can be held responsible for them. The funders had no role in study design, data collection and analysis, decision to publish or preparation of the manuscript.

## Author contributions

W.D.N. conceived the project and acquired funding. W.D.N. and M.R. designed the experiments. M.R. implemented the study design, collected the data, and analyzed all data. W.D.N. and M.R. drafted the paper and provided revisions. Both authors approved the final paper for submission.

## Competing interests

The authors declare no competing interests.
