## [Transparent Peer Review file · Communications Psychology]

HUMANS AND LLMs RATE DELIBERATION AS SUPERIOR TO INTUITION ON COMPLEX REASONING TASKS

Corresponding Author: Professor Wim De Neys

Version 0:

Decision Letter:

Dear Professor De Neys,

Thank you for submitting your manuscript titled "Folk thinking, fast and slow: intuitive preference for deliberation in humans and machines" to Communications Psychology. We have given the paper our careful consideration and find it of potential interest. We have also read the reviews received at Nature Human Behaviour. The reviewers found your work of interest but raised some important points. We are interested in the possibility of publishing your study in Communications Psychology, but would like to consider your responses to these concerns and assess a revised manuscript before we make a final decision on publication.

We therefore invite you to revise and resubmit your manuscript, along with a point-by-point response to the reviewers. Please highlight all changes in the manuscript text file.

Editorially, we consider it important the revision acknowledges the scope and limitations of the current work regarding domain restrictions, measurement of the dependent variable, and ecological validity.

Please ensure you follow our statistical guidelines when reporting statistics (<https://www.nature.com/commpsychol/submit/submission-guidelines#statistical-guidelines>). Please note in particular our requirements for the reporting and interpretation of null-results. Non-significant findings derived from null-hypotheses significance tests should be reported in full, but may not be interpreted. Where you interpret null results, this interpretation must be based on Bayes Factors or equivalence tests. Please add these for studies 10-13.

I am attaching an Editorial Requests Table that details critical reporting requirements for the revised manuscript. Please attend to each item and ensure your manuscript is fully compliant. If your revised manuscript is not aligned with these requests on major issues, such as those concerning statistics, it may be returned to you for further revisions without re-review.

Please submit the following items:

- Revised manuscript
- Point-by-point response to the referees' comments
- Cover letter (as a separate document)
- <https://www.nature.com/documents/nr-reporting-summary.zip>>Nature Research Reporting Summary
- <https://www.nature.com/documents/nr-editorial-policy-checklist.pdf>>Editorial Policy Checklist

- Completed Editorial Request Table (attached).

via this link: Link Redacted .

Additional guidance is available in our style and formatting guide Communications Psychology formatting guide.

Best regards,

Jennifer Bellingtier

Jennifer Bellingtier, PhD
Senior Editor
Communications Psychology

If you experience problems in linking your ORCID, please contact the Platform Support Helpdesk.

Version 1:

Decision Letter:

Dear Professor De Neys,

Your manuscript titled "Folk thinking, fast and slow: intuitive preference for deliberation in humans and machines" has now been seen by our reviewers, whose comments appear below. In light of their advice I am delighted to say that we are happy, in principle, to publish a suitably revised version in Communications Psychology.

We therefore invite you to revise your paper one last time to address the remaining concerns of our reviewers and a list of editorial requests. At the same time we ask that you edit your manuscript to comply with our format requirements and to maximise the accessibility and therefore the impact of your work.

EDITORIAL REQUESTS:

SUBMISSION INFORMATION:

OPEN ACCESS:

* DATA AVAILABILITY:

Link Redacted

Best regards,

Jennifer Bellingtier

Jennifer Bellingtier, PhD
Senior Editor
Communications Psychology

REVIEWERS' COMMENTS:

Reviewer #2 (Remarks to the Author):

I was an original reviewer on this manuscript. I continue to find the topic timely and important. I appreciate several of the revisions that were made, although my reservations about how much we learn from the LLM results and the null effects of load / time pressure remain. I have two small suggestions for revisions:

1. The authors have made some revisions to address one of my primary concerns, which concerned the scope of their results, which are likely to depend very much on the decision task used. I appreciate the changes but I think more could be done. Currently, the authors contrast “subjective choices” with “cognitively challenging tasks”, where their studies focus on the latter. At least for me this isn't a clear contrast: I would classify many subjective choices as also cognitively challenging. Perhaps in the introduction the authors can be more explicit about what they do focus on (complex reasoning tasks with objectively correct responses?), and say something about the motivation for this focus.
2. In the introduction, the authors write (referring to references 14-17): “this early work conflates thinking style and accuracy”. The authors make a valid point in their rebuttal: That accuracy in a given case is not the same as historical accuracy across many cases (which I will refer to as reliability). But this prior work does look at aspects of both accuracy and reliability. For example, Otkar & Lombrozo control for accuracy in a given case (which I acknowledge is not the same as historical accuracy /reliability), but also measure and control for the perceived reliability of a process for a given domain. The authors could revise this paragraph to better reflect past work and the novelty of their contribution.

Reviewer #3 (Remarks to the Author):

The authors answered all my concerns and provided well-motivated responses. On my side, there are no further aspects to clarify.

In cases where reviewers are anonymous, credit should be given to 'Anonymous Referee' and the source. The images or other third party material in this Peer Review File are included in the article's Creative Commons license,

Dear Editor,

I am pleased to submit our revised manuscript, "Folk Thinking, Fast and Slow: Intuitive Preference for Deliberation in Humans and Machines," for consideration at Communications Psychology. I am attaching a detailed response to the reviewer comments from Nature Human Behaviour. All revisions are highlighted in red.

Sincerely,

Wim De Neys
Sorbonne – LaPsyDE
Université Paris Cité
46, rue Saint Jaques
75005 Paris
France

Editor

Editorially, we consider it important the revision acknowledges the scope and limitations of the current work regarding domain restrictions, measurement of the dependent variable, and ecological validity.

Response: We added an explicit limitations section and further acknowledged the limitations of the current work in line with the reviewer suggestions.

Please ensure you follow our statistical guidelines when reporting statistics (<https://www.nature.com/commpsychol/submit/submission-guidelines#statistical-guidelines>).

Please note in particular our requirements for the reporting and interpretation of null-results. Non-significant findings derived from null-hypotheses significance tests should be reported in full, but may not be interpreted. Where you interpret null results, this interpretation must be based on Bayes Factors or equivalence tests. Please add these for studies 10-13.

Response: We have added the Bayes Factors analysis but see also our reply to Reviewer 2.

Reviewer #1 (Remarks to the Author):

This manuscript presents a compelling series of studies examining lay perceptions of intuitive versus deliberative reasoning in both humans and large language models. The authors report a robust and consistent preference for deliberative reasoners/reasoning, independent of actual accuracy. Notably, this preference persists across diverse populations and under cognitive constraints, suggesting that the preference for deliberators may itself be intuitive. Participants not only rated deliberative thinkers as better reasoners but also perceived them as more intelligent and trustworthy. What I find particularly interesting and valuable is how LLMs mirrored human preferences (S8 y9), suggesting that this preference is somehow built in our language. This also in turn raises exciting questions for how AI-generated content is evaluated and trusted by users.

Response: Thanks!

This last aspect though is also the one that raises my main concern. How is the evaluation for preference for deliberate reasoning independent from the very concept of deliberate reasoning? The more general question is about the concept of deliberation/deliberator that allows reasoners both people and large language models to establish the preference and how this preference is independent from the concept of deliberation/deliberator itself. Let's consider the Dv's. The scales used in most of the studies (1 to 4 and 6 to 10) assess how good, smart and trustworthy the person is at reasoning ("How good is this person at reasoning?"; "How smart is this person?"; "To what extent would you follow this person's advice about a reasoning problem?") or overall aptness at reasoning ("How good is this person at reasoning" in Studies 5 and 11-13). In the ranking study instructions (S2, 10 and 11), people were told "You will have to rank people from the most intelligent (number 1) to the least intelligent (number 4).

First of all, although the paper goes back and forth between the claim that people evaluated the reasoning processes and the reasoners, it is clear that results refer to the person in all cases. This is a subtle but key difference, since we do not know if there is a dissociation between act and person, as

it is found in areas like moral psychology (e.g. a deliberate person might sometimes make intuitive decisions, etc). This is a minor issue.

Response: Our studies asked participants to evaluate the inferences of a person who used a specific reasoning style (intuition vs. deliberation). The wider issue the reviewer (as well as Reviewer 3) refers to is whether these styles can also diverge—can people switch from one style to another? Put differently, should reasoning style be considered a stable trait or a flexible, context-dependent process? While this is certainly an interesting question for future research, it is orthogonal to our central aim. Our focus is on how people evaluate a reasoner who used a particular style, irrespective of whether that style is seen as fixed or malleable. Across 13 studies, we observe a robust, intuitive preference for deliberation. The belief that reasoning styles may shift over time does not change how people evaluate a decision-maker who employed a specific approach. But we agree it is a good idea to note this point more explicitly. We now discuss it in our limitation section and suggest it as one avenue for follow-up work. This is what we write:

“Third, our studies asked participants to evaluate the inferences of a person who used a specific reasoning style (intuition vs. deliberation). A further question is whether people also believe these styles can shift—do they think that a reasoner can switch from one style to another? Put differently, is reasoning style seen as a stable trait or as something more flexible and context-dependent? Our studies did not address this issue. Our goal was to assess how people evaluate a reasoner based on the style used in a specific moment. The belief that reasoning styles may shift over time does not undermine such evaluations. Even if a person might reason differently in the future, people can still judge the quality of their reasoning in the situation at hand. Nevertheless, although the malleability question was orthogonal to our central aim, it presents an interesting avenue for future research.”

“Intelligent” and “smart” are used to evaluate preference for deliberate reasoners. I think it is fair to ask whether deliberation is one of the defining features of being “Intelligent” and “smart”. Intuitively (pun intended), this might be the case for at least some communities or at the very least one could expect variation on this aspect. “Good at reasoning” might also more be tightly connected with being a “deliberate reasoner” at a conceptual level than to being an “intuitive reasoner”. Isn’t intelligence defined partly by “deliberating” but not by intuition?. I know it is for me, but it is clearly a more general question on the folk concept of intuitive and deliberate reasoners.

One way of responding to this issue is by showing that the reliability of the three indicators used in most studies is high, as it is actually presented in the paper. However, what I believe is needed is a more principled way to distinguish the concept evaluated from the aspects used to evaluate. High reliability might have been reached for the wrong reasons (e.e. Positivity bias) and it more generally does not guarantee content validity. Maybe the process that allowed the authors to settle for the DV’s used will be illuminating on this point.

Response: We agree that one may use a wide range of labels to characterize reasoning quality. We adopted 3 different common questions (“how good at reasoning?”, “how smart?”, “how likely to take advice?”) that have been used previously to measure decision quality (e.g., Castello et al., 2019; Kupor et al., 2014). We observed robust, high correlations among them. Moreover, our studies that adopted a single rating scale showed completely similar results. This indicates that our results are

not driven by any specific question label. However, we agree it is a good idea to specify this methodological point more explicitly. In the revision we write:

“For each vignette, participants rated the quality of the described individual’s reasoning by clicking on the corresponding value on an 11-point rating scale (0-10). Except for Studies 5 and 11-13, we presented three different scales (“How good is this person at reasoning?”; “How smart is this person?”; “To what extent would you follow this person’s advice about a reasoning problem?”). The scale labels were inspired by previous work measuring decision quality (e.g., Castello et al., 2019; Kupor et al., 2014). The three scales were presented simultaneously, directly under the vignette on one single page. ...”

I'm aware that conceptual distinctions between what is evaluated and how you evaluate it is open to discussion but is linked to an issue explicitly acknowledged by the authors but not sufficiently explored: ecological validity. I do not understand how vignettes like the ones used in the studies (“Person A follows their intuition when reasoning about a problem. They do not spend much time or effort to arrive at a conclusion. The accuracy of their answers is very high.”) were “designed to depict situations where individuals encounter complex reasoning tasks” (p. 9) in the absence of any context. Participants have to bring in too much information to the inference task at hand. Presumably the information they bring with them relates to the very definition of intuitive and deliberate reasoner, making it even more difficult to generalize from these vignettes to real world settings. There seems to be just too many aspects of people problems and situations that would make preference for deliberate reasoners a lot more nuanced than what is presented.

Response: We agree that one always needs to remain cautious when generalizing from vignette studies. However, our results do suggest that our findings are not driven by any specific idiosyncratic feature of our specific vignettes. In our validation studies (Study 2-7) we used different phrasings and measurements (3 vs 1-scale) and found similar results as in Study 1. We also showed that vignette choice preferences were consequential (betting paradigm) and robustly observed in different cultures. Moreover, the fact that LLMs showed identical preferences indicate that the vignettes match with common descriptions of intuition and deliberation in our daily language and communication. Nevertheless, we agree that vignettes, though controlled, do not capture the full ecological complexity of real-world judgments and remain to be interpreted with caution. This is what we write in the limitations section:

Second, while our vignette-based approach allowed us to isolate reasoning mode as a variable, it also comes with limitations. Vignettes, though controlled, do not capture the full ecological complexity of real-world judgments, where factors like social context and personal experience could influence reasoning preferences. Moreover, although our incentivized paradigm suggests a genuine preference for deliberation over intuition, future research could confirm these findings in more interactive settings. For instance, if individuals repeatedly interact with a person and learn from these experiences, their initial preference, based on a single descriptive vignette, might shift. Clearly, given that people often make trust decisions on initial impressions, especially in first-time interactions, our approach still captures an essential aspect of real-world judgment. At the same time, it would be valuable to further explore how preferences for deliberation evolve with ongoing experience and familiarity in real-world settings.

We repeated the limitation in our conclusion:

*“In conclusion, our findings reveal a robust, intuitive preference for deliberative reasoning that is echoed in AI models like LLMs. This highlights an implicit belief associating deliberation with reliable, thoughtful reasoning, carrying implications for fostering public trust while also presenting risks of exploitation. **At the same time, the findings are shaped by certain boundary conditions—including our focus on cognitively challenging tasks, vignette-based methods, and relatively educated samples. Future work will be needed to explore how robust these preferences are across different judgment domains, real-world settings, and more diverse populations.**”*

It would be great to see a response to these points. In any case, I believe this is a very valuable work and I particularly insist on the very interesting finding of LLMs mirroring human inference patterns in the absence of the context pointed out.

Other

Reference 20 is incorrect, could not find out what it refers to.

Response: corrected.

Reviewer #2 (Remarks to the Author):

This paper reports several experiments investigating people's preference for a problem solver who relies on deliberation or intuition in reaching a conclusion. Studies 1-7 find a consistent preference for the deliberative problem solver, despite variation in vignette wording, different samples, the addition of incentives in one study, and other small changes. Studies 8-9 find that LLMs show the same preference. Studies 10-13 find that the preference persists under time pressure or cognitive load.

Overall this paper has a lot of really nice features. It is clearly written, and includes many internal replications and variants that help bolster the shared finding across studies 1-7. I think the topic is important, of general interest, and potentially timely.

Despite these strengths, I do have some reservations about the way the findings are described, and about whether the contribution meets what I take to be the very high bar for this journal. I will consider the paper's introduction and then sets of experiments in turn.

Response: Thanks! We reply to your specific points and the more general, underlying point about domain-specificity below.

introduction

The introduction is very clear and does a nice job motivating the topic. However, I think it simplifies the literature in a way that is misleading. At the bottom of page 3, we are told that intuition is in fact more predictive of accuracy than deliberation. My understanding is that this depends enormously on the nature of the problem, and so it is misleading as an unqualified generalization. Similarly, the authors refer to "inherent" algorithm aversion (page 4), which makes it sound like this is a robust and domain-general aversion, but it is not: it depends on the decision domain and other features of the task. There are several relevant references, but here are two:

Castelo, N., Bos, M. W., & Lehmann, D. R. (2019). Task-dependent algorithm aversion. *Journal of marketing research*, 56(5), 809-825.

Logg, J. M., Minson, J. A., & Moore, D. A. (2019). Algorithm appreciation: People prefer algorithmic to human judgment. *Organizational Behavior and Human Decision Processes*, 151, 90-103.

Response: We did not want to imply that algorithm aversion is robust or unavoidable. Our choice of the adjective "inherent" was unfortunate in this respect. We have replaced it with "possible".

With regard to prior work on the main topic of the paper, four main papers come to mind:

Inbar, Y., Cone, J., & Gilovich, T. (2010). People's intuitions about intuitive insight and intuitive choice. *Journal of personality and social psychology*, 99(2), 232.

Kupor, D. M., Tormala, Z. L., Norton, M. I., & Rucker, D. D. (2014). Thought calibration: How thinking just the right amount increases one's influence and appeal. *Social Psychological and Personality Science*, 5(3), 263-270.

Oktar, K., & Lombrozo, T. (2022). Deciding to be authentic: Intuition is favored over deliberation when authenticity matters. *Cognition*, 223, 105021.

Pachur, T., & Spaar, M. (2015). Domain-specific preferences for intuition and deliberation in decision making. *Journal of Applied Research in Memory and Cognition*, 4(3), 303-311.

I only saw the Kupor and Oktar cited. Also closely related (and not cited) is the following paper on decision speed (which is plausibly a proxy for intuitive versus deliberative decision making)

Critcher, C. R., Inbar, Y., & Pizarro, D. A. (2013). How quick decisions illuminate moral character. *Social Psychological and Personality Science*, 4(3), 308-315.

I think the introduction / literature review should be amended to include these papers.

Response: Note that we did cite and discuss both Kupor et al. (2014) and Oktar et al. (2022). We had missed the older Inbar et al. (2010) paper, which we have now incorporated.

Experiments 1-7

A consistent finding from the previous work on preferences for intuitive and deliberative decision making is that the preference depends on the decision domain. So it matters enormously what the nature of the decision is. We don't get a sense for this until the materials section, but the decisions all involve unspecified problems that have objective solutions, since accuracy is specified. These are conditions under which we would expect, from all previous work, to find a preference for deliberation. I think the restriction to certain kinds of problems needs to be made clear at the outset (currently it is acknowledged in the discussion, and we learn about the materials in the methods). This also means that claims need to be qualified – the findings apply to these kinds of problems, but looking at the literature as a whole, we should not expect a general preference for deliberation. Given the prior work, I think the most novel contribution of these studies is the incentivized study, which finds that a preference for deliberation persists (and is perhaps magnified) when participants are incentivized.

Response: Reviewer 2's main issue is that reasoning preferences may depend on task domain or features. We agree, and we explicitly acknowledged this point in the general discussion. Our studies focused on how people evaluate reasoning styles in cognitively challenging, general-purpose problems. We ran 13 studies to robustly establish the nature of these broad preferences. Of course, preferences may shift in non-cognitive domains (e.g., romantic decisions) or in areas where individuals are experts. We noted this in the manuscript and agree it is an important direction for follow-up research. In the revision we have made this clearer by integrating it in the limitations section. Following the Communications Psychology guidelines, we also put the method section before the results section which should help to contextualize the results. Finally, following the

reviewer's suggestion we noted the restriction to certain types of problems directly in the introduction. This is what we write in the introduction:

Although people may prefer intuition when making subjective decisions (e.g., whom to date; see Inbar et al., 2010; Oktar & Lombrozo, 2022), initial evidence suggests that humans—even from an early age (Richardson & Keil, 2022)—tend to favor deliberation over intuition when faced with cognitively challenging tasks (Kupor et al., 2014; Oktar & Lombrozo, 2022). However, this early work conflates thinking style and accuracy. People may prefer deliberation because they believe it will be more accurate. But what if we know that the decision maker is an experienced expert who is highly accurate? This question is particularly relevant given that celebrated cases of intuitive success—from Jobs to Sullenberger to Gladwell's examples—typically showcase experts who are consistently accurate. So, what do we prefer if two individuals are both highly accurate? Do we trust the individual who intuitively “sees” the right solution more than the one who needs to spend time and effort to arrive at an answer?

Discussion:

“Limitations

When considering the broader implications of our findings, it is essential to acknowledge certain limitations of our studies. First, we focused specifically on folk beliefs about the role of intuition and deliberation in addressing cognitively challenging problems. Our vignettes were designed to depict situations where individuals encounter complex reasoning tasks. In trivial or straightforward scenarios (e.g., answering, “How much is 5 + 5?”), people are unlikely to favor a reasoner who needs slow, painstaking deliberation; in such cases, the need to deliberate could signal cognitive limitations rather than sophistication [15]. Similarly, for non-cognitive or subjective judgments (e.g., a partner asking, “Do you love me?”), people may prefer a swift, intuitive response, which is often seen as more authentic [16, 36, 37]. Thus, it is improbable that our findings extend to (AI) recommendations for highly subjective areas like romantic choices or personal preferences in art or music, for example. More generally, it would be beneficial to investigate the generalizability of these preferences across diverse judgment domains.”

And conclusion:

*“In conclusion, our findings reveal a robust, intuitive preference for deliberative reasoning that is echoed in AI models like LLMs. This highlights an implicit belief associating deliberation with reliable, thoughtful reasoning, carrying implications for fostering public trust while also presenting risks of exploitation. **At the same time, the findings are shaped by certain boundary conditions—including our focus on cognitively challenging tasks, vignette-based methods, and relatively educated samples. Future work will be needed to explore how robust these preferences are across different judgment domains, real-world settings, and more diverse populations.**”*

Another claim of novelty is made in the general discussion: “Our findings refine these insights [that people favor deliberation for complex tasks] by demonstrating that people’s appreciation for deliberation is largely intuitive and independent of accuracy.” Oktar & Lombrozo also includes studies that control for accuracy.

Response: This is perhaps a minor issue but here we want to correct the reviewer. The reviewer suggests that our accuracy manipulation is less novel than claimed, arguing that Oktar et al. (2022) similarly controlled for accuracy. However, this is not the case. We provided participants with explicit information about the decision-maker's past accuracy (high, low, or unspecified), whereas Oktar et al. manipulated whether intuitive and deliberative processes led to the same conclusion. In dual-process research, process concordance is not equivalent to demonstrated accuracy. It is not because an intuitive and deliberate process converge, that the resulting outcome is therefore valid (e.g., in the bat-and-ball problem many reasoners will both intuitively and after deliberation claim that the correct answer is "10 cents"). Our manipulation was designed specifically to disentangle reasoning style from performance, and is therefore critical for drawing the conclusions we report. It is indeed novel and original in this respect.

Experiments 8-9

These LLM studies suggest that LLMs mirror human judgments. What do we learn from this? The authors write that it is "likely because these patterns [of human preference for intuition / deliberation] are well-represented in the language data used to train the models." That seems very plausible to me, but I don't know that my belief in the claim is changed substantially by the actual LLM results. Without some deeper analysis here of which language patterns are relevant, or how / why LLMs and humans are similar or differ in some ways, I didn't get a lot of insight from these results.

Response: The finding that LLMs mirror human preferences indeed indicates that the preference for deliberative reasoning may be a widely shared belief evident in common language and communication data (on which the models were trained). As Reviewer 1 and 3, we find this remarkable. It does suggest that we are looking at a very robust, basic preference.

Experiments 10-13

My main concern here is that the conclusions rest on a null result: that a speeded / high load condition did not differ from an unspeeded condition. These results would be more compelling with some matched control task that did show effects.

Response: Yes, some of the ANOVA results involving the restriction factor and its interactions yielded null effects. However, it is important to emphasize that our manipulation employed some of the most stringent conditions used in the literature to date—combining instruction constraints, time pressure, and cognitive load. Each of these techniques has individually been shown to reliably restrict deliberation, so the fact that we consistently observed null effects across four studies is far from trivial.

That said, our central claim does not rest on the absence of restriction effects in the ANOVA analyses per se. Rather, the key finding is that, across all restriction conditions, participants showed a significant pairwise preference for deliberation over intuition, regardless of accuracy. This demonstrates that the deliberation preference is robust across all accuracy and restriction manipulations.

In line with the journal's guidelines, we now also report Bayes Factor analyses. These indicate that support for the null hypothesis concerning the restriction effects is relatively weak ($0 \leq BF_{01} \leq 5.88$)—a point we now acknowledge. At the same time, the Bayes Factors for the pairwise contrasts between intuitive and deliberative reasoning style consistently show very strong evidence in favor of a deliberation preference across all restricted conditions ($1.1 \times 10^4 < BF_{10} < 1.22 \times 10^5$). This further supports our conclusion that the preference for deliberation does not itself require deliberative processing. We made sure to clarify this in the revision. This is what we write:

“For clarity, we should stress that—as indicated above—some of the ANOVA results involving the restriction factor and its interactions yielded null effects. In line with the journal's guidelines and reviewer suggestions, we also ran Bayes Factor analyses. These indicate that support for the null hypothesis concerning the restriction effects is relatively weak ($0 \leq BF_{01} \leq 5.88$, see SI L). That said, our central claim does not rest on the absence of restriction effects in the ANOVA analyses per se. Rather, the key finding is that, across all restriction conditions, participants showed a significant pairwise preference for deliberation over intuition, regardless of accuracy. This indicates that the deliberation preference is robust across our accuracy and restriction manipulations. For completeness, we also ran a Bayes Factors analysis for the pairwise contrasts between intuitive and deliberative reasoning mode vignettes. Results consistently showed extreme evidence in favor of a deliberation preference across all restricted conditions ($1.1 \times 10^4 < BF_{10} < 1.22 \times 10^5$, see SI L). This further supports the conclusion that the preference for deliberation does not itself require deliberative processing.”

In sum, this is a nice paper that contributes to a small but consistent literature finding preferences for deliberative reasoning *for some kinds of problems.* I think the literature review and conclusions need to be more explicit about this qualification, and about the limitations of the current studies, given that they focus exclusively on relatively abstract / underspecified objective problems. I think the paper's most novel contribution is the study that includes incentives for accuracy. I find the LLM results undeveloped / relatively uninformative on their own, and the speeded / load tasks only partially convincing. With revisions, I think this paper should be published. But I'm not sure it meets the very high bar for *this* journal.

Reviewer #3 (Remarks to the Author):

In this paper, the authors investigate whether participants prefer a more intuitive or deliberative process when solving problems. In the experiments, participants are presented with vignettes specifying the type of reasoning (intuitive or deliberative) and the level of accuracy of the answers (high, low or absent). Moreover, the authors explore whether Large Language Models (LLMs) favor intuition or deliberation in the same tasks proposed to humans. Overall, the authors find that participants prefer deliberation over intuition, and this remains true even when accuracy is considered. LLMs show similar results to those observed in humans.

I find the research question important, and this manuscript is interesting. The paper is well written, and its message is clear.

Response: Thanks!

Below, I list my main concerns and suggestions:

The topic is of general interest, and it would be interesting to expand the section of the introduction by discussing more the policy implications of folk thinking, for example how preferences for one type of reasoning over another may influence individuals' decision-making. While this part is already present in the discussion section, I suggest also adding it in the introduction.

Response: given that the policy implications remain speculative we prefer to keep them in the General Discussion.

Moving to the study design, I have some concerns. I would better understand the choice of the questions asked to the participants. Participants are asked the following questions: "How good is this person at reasoning?", "How smart is this person?" and "To what extent would you follow this person's advice about reasoning problem". Classify how good the person is at reasoning and how smart the person is without knowing anything but just the type of reasoning appears strange to me. How do these questions link to the preference for a type of reasoning? It seems that they assess participants' perceptions of the person rather than their evaluation of the reasoning style itself. I would like to be convinced by the authors about this choice.

Response: Our studies asked participants to evaluate the inferences of a person who used a specific reasoning style (intuition vs. deliberation). The wider issue the reviewer refers to is whether these styles can also diverge—can people switch from one style to another? Put differently, should reasoning style be considered a stable trait or a flexible, context-dependent process? While this is certainly an interesting question for future research, it is orthogonal to our central aim. Our focus is on how people evaluate a reasoner who used a particular style, irrespective of whether that style is seen as fixed or malleable. Across 13 studies, we observe a robust, intuitive preference for deliberation. The belief that reasoning styles may shift over time does not change how people evaluate a decision-maker who employed a specific approach. But we agree it is a good idea to note this point more explicitly. We now discuss it in our limitation section and suggest it as one avenue for follow-up work. This is what we write:

“Third, our studies asked participants to evaluate the inferences of a person who used a specific reasoning style (intuition vs. deliberation). A further question is whether people also believe these styles can shift—do they think that a reasoner can switch from one style to another? Put differently, is reasoning style seen as a stable trait or as something more flexible and context-dependent? Our studies did not address this issue. Our goal was to assess how people evaluate a reasoner based on the style used in a specific moment. The belief that reasoning styles may shift over time does not undermine such evaluations. Even if a person might reason differently in the future, people can still judge the quality of their reasoning in the situation at hand. Nevertheless, although the malleability question was orthogonal to our central aim, it presents an interesting avenue for future research.”

Following on the questions asked to participants, do the authors also conduct separate analyses or regressions for each question?

Response: Yes, we also conducted separate analyses for each question. These are reported in the Supplementary Material. As we noted, results were fully consistent with the composite pattern.

Moreover, it would be interesting to see if results are stable when controlling for gender, age, level of education (which were collected on the study) and other variables, if any.

Response: We did not have any hypotheses concerning these variables and possible individual differences and prefer to not report any exploratory findings to avoid spurious conclusions. However, our data are fully open and anyone with an interest in these demographic variables can access and explore them. We personally feel that looking at these and other possible individual differences factors (e.g., link with IQ, personality factors, etc.) should be the object of a direct preregistered study.

On this point I was wondering if authors control for attention check.

Response: Given the possible backlash of attention check questions we never use them in our lab. However, if participants were generally inattentive and guessing in the present studies, we should not observe the systematic, robust preference for deliberation.

I have concerns about the incentivized part (the betting task), specifically regarding the use of deception. I am generally skeptical when we talk about deception. Why was deception necessary in this case? I think it is generally good to avoid deception whenever possible unless there is a strong motivation for its use.

Response: Point taken. Because the vignettes were hypothetical we could not link them to the performance of a real individual. We consequently opted for deception but awarded the same bonus to all participants.

I appreciate and I find interesting the studies on LLMs.

Response: Thanks!

Finally, I suggest stating the number of participants in each study, as well as where participants were recruited, in the results sections; currently this information is in the method section.

Response: Corrected. Following the journal guidelines, we now have the method section before the results section.

Dear Editor,

I am pleased to submit our revised manuscript for consideration at Communications Psychology. I am attaching a response to the few remaining reviewer comments. All revisions are highlighted in red.

Sincerely,

Wim De Neys
Sorbonne – LaPsyDE
Université Paris Cité
46, rue Saint Jacques
75005 Paris
France

I was an original reviewer on this manuscript. I continue to find the topic timely and important. I appreciate several of the revisions that were made, although my reservations about how much we learn from the LLM results and the null effects of load / time pressure remain. I have two small suggestions for revisions:

1. The authors have made some revisions to address one of my primary concerns, which concerned the scope of their results, which are likely to depend very much on the decision task used. I appreciate the changes but I think more could be done. Currently, the authors contrast “subjective choices” with “cognitively challenging tasks”, where their studies focus on the latter. At least for me this isn’t a clear contrast: I would classify many subjective choices as also cognitively challenging. Perhaps in the introduction the authors can be more explicit about what they do focus on (complex reasoning tasks with objectively correct responses?), and say something about the motivation for this focus.

Response: Ok, we have phrased this more clearly. We now write:

“Although people may prefer intuition when making decisions about subjective choices (e.g., what partner to date, e.g., [14, 15]), some initial work does seem to suggest that humans—even from a young age [16]—generally tend to prefer deliberation over intuition when facing **more objective, cognitively challenging reasoning tasks [14, 17].”**

Note that the next paragraph also clarifies the point. In addition, we opted for the suggested revised title “HUMANS AND LLMs RATE DELIBERATION AS SUPERIOR TO INTUITION ON COMPLEX REASONING TASKS**” that makes our specific focus crisp clear from the outset.**

2. In the introduction, the authors write (referring to references 14-17): “this early work conflates thinking style and accuracy”. The authors make a valid point in their rebuttal: That accuracy in a given case is not the same as historical accuracy across many cases (which I will refer to as reliability). But this prior work does look at aspects of both accuracy and reliability. For example, Otkar & Lombrozo control for accuracy in a given case (which I acknowledge is not the same as historical accuracy /reliability), but also measure and control for the perceived reliability of a process for a given domain. The authors could revise this paragraph to better reflect past work and the novelty of their contribution.

Response: Yes, but our original point still stands, the O&L study did not control for accuracy in the critical sense we define and need it here. Anyway, to avoid further discussion we have phrased the claim more cautiously and noted that “earlier work **often conflates thinking style and accuracy”. We believe we can all agree that this is factually correct.**